# Exploring the Suicide Mechanism Path of High-Suicide-Risk Adolescents—Based on Weibo Text Analysis

**DOI:** 10.3390/ijerph191811495

**Published:** 2022-09-13

**Authors:** Liuling Mo, He Li, Tingshao Zhu

**Affiliations:** 1Institute of Psychology, Chinese Academy of Sciences, Beijing 100101, China; 2Department of Psychology, University of Chinese Academy of Sciences, Beijing 100049, China

**Keywords:** high-suicide-risk adolescents, psychological pain, hopelessness, suicide stages, Weibo text analysis

## Abstract

Background: Adolescent suicide can have serious consequences for individuals, families and society, so we should pay attention to it. As social media becomes a platform for adolescents to share their daily lives and express their emotions, online identification and intervention of adolescent suicide problems become possible. In order to find the suicide mechanism path of high-suicide-risk adolescents, we explore the factors that influence is, especially the relations between psychological pain, hopelessness and suicide stages. Methods: We identified high-suicide-risk adolescents through machine learning model identification and manual identification, and used the Weibo text analysis method to explore the suicide mechanism path of high-suicide-risk adolescents. Results: Qualitative analysis showed that 36.2% of high-suicide-risk adolescents suffered from mental illness, and depression accounted for 76.3% of all mental illnesses. The mediating effect analysis showed that hopelessness played a complete mediating role between psychological pain and suicide stages. In addition, hopelessness was significantly negatively correlated with suicide stages. Conclusion: mental illness (especially depression) in high-suicide-risk adolescents is closely related to suicide stages, the later the suicide stage, the higher the diagnosis rate of mental illness. The suicide mechanism path in high-suicide-risk adolescents is: psychological pain→ hopelessness → suicide stages, indicating that psychological pain mainly affects suicide risk through hopelessness. Adolescents who are later in the suicide stages have fewer expressions of hopelessness in the traditional sense.

## 1. Introduction

Adolescent suicide has always been a global public health concern to us. Suicide is the fourth leading cause of death globally among people aged 15–19 [1], and in China, it is the second leading cause of death for people aged 20–34 [2]. The consequences of adolescent suicide are the most serious among many mental health problems. It means disability or even a loss of life for individuals, heavy blows and severe trauma for families, and huge expenditures on the health, welfare and justice sectors for society. The study of suicide risk factors and the exploration of suicide mechanisms in high-suicide-risk adolescents are of great significance for the prevention of suicide.

With the development of technology, social media has become a platform for people to record their daily life and express their emotions. In China, Sina Weibo as a typical product of the big data era has extensive influence. Sina Weibo is China’s most popular social media platform, just like Twitter in America. Although there are some short video platforms (e.g., Douyin) and forums (e.g., Douban) that are also popular in China, it is only on Sina Weibo that users can share information in real-time and communicate interactively in the forms of text, pictures, videos and other multimedia. According to a report released by Sina Weibo, Weibo users show a younger trend, of which the post-90s and post-00s youth groups account for nearly 80%, and the daily active volume can reach 224 million [3].

The user’s emotional expression on social platforms has higher immediacy and less masking. Identifying the user’s psychological state through their expression on social media will be more convenient and quicker, and the obtained data will be more real. Therefore, we use the Weibo text analysis method to explore the suicide mechanism path of high-suicide-risk adolescents. Text analysis refers to the representation of text and the selection of feature items, which is a basic problem in text mining and information retrieval. Text analysis means that we can quantify the feature words extracted from text to represent text information. Therefore, we can extract the psychological indicators we want from the Weibo text by constructing a dictionary of psychological indicators.

Suicide represents a series of consecutive stages, including suicide ideation, suicide plan, and suicide attempt [4,5]. Suicide ideation, as a high-risk factor for suicide, has a significant predictive effect [6,7]. Approximately 1/3 of suicide ideators ultimately attempt suicide, and the number of suicide attempters is several times higher than those who really die by suicide, so suicide attempt is a greater risk factor for suicide than suicidal ideation [1,8,9]. The later the suicide stage, the closer to suicide death, and the greater risk of suicide. Therefore, in order to quantify the degree of suicide risk and provide accurate information for suicide interventions, we decided to divide suicidal behavior into different stages. At the same time, considering that depressed mood is an important signal of suicide, we finally divided suicide into 4 stages, which the suicide risk gradually increases: depressed mood, suicide ideation, suicide plan and suicide attempt, and defined adolescents at different suicide stages as high-suicide-risk adolescents.

A meta-analysis of the causes of suicide among adolescents found that suicide is a public health problem involving complex factors, such as biology, psychology, family, society and culture [8]. Among the psychological factors, psychological pain and hopelessness are important risk factors featuring strongly in suicide research [9,10,11]. Although psychological pain and hopelessness are often mentioned together as having an impact on suicidal behavior, as two most common motivations for suicide attempts [12], it is unclear how the two proximal factors of suicide affect the suicide stages.

Psychological pain is defined as the “introspective experience of negative emotions such as fear, despair, grief, shame, guilt, blocked love, loneliness and loss” [13,14,15,16]. When Shneidman first hypothesized the relationship between psychological pain and suicide, he pointed out that other factors can only affect suicidal behavior through psychological pain, without which suicide would not occur [13]. Evidence suggests that psychological pain plays a central role in suicidal behavior [10,17,18]. Psychological pain is not enough to develop suicide ideation, it also requires hopelessness [19]. Hopelessness is defined as negative expectations or pessimism about oneself or the future [20]. Previous studies had showed that hopelessness was also an important risk factor of suicide [21,22,23].

To explore the relationship between psychological pain, hopelessness and suicide stages for high-suicide-risk adolescents, we established a hypothesis for the suicide mechanism path: psychological pain → hopelessness → suicide stages, with hopelessness as the mediating variable. In addition, there is a high correlation between mental illness and suicidal behaviors [24,25], and in order to better clarify the effect of psychological pain and hopelessness on suicide stages, a mediating effect model of our study is proposed with mental illness as a control variable (Figure 1). We used the more ecological Weibo text analysis method in this study, and hoped to further deepen the research on adolescent suicide and to find a new theoretical basis for future suicide interventions drawing on online data.

## 2. Materials and Methods

### 2.1. Participants

Through the online psychological assistance project, we obtained the interview records data of 3627 Weibo users, and then processed the user’s interview records data and Weibo texts data. Finally, we retained 105 high-suicide-risk adolescents with enough Weibo texts as the experimental group.

Ethical Statement: Participants gave informed consent when agreeing to take part in the online psychological assistance project. The project received ethical approval from the Institution-al Review Board of the Institute of Psychology, Chinese Academy of Sciences, with the ethics approval number H16003.

### 2.2. Online Psychological Assistance Project

We used the suicide signal recognition model established by the Computational Cyber-Psychology Lab (CCPL) to identify the suicide signal of a single microblog, which can effectively determine whether the single microblog has shown suicidal risk [26].

Since November 2016, we had used the suicide signal recognition model for online identification of Weibo comments. The last Weibo post before the death of Weibo user “Zoufan”, as a place where people can express their negative emotions, has received more than 1 million comments until now, and the number of comments keep rising (Figure 2).

Firstly, we used the suicide signal model to identify the suicide signal of the Weibo comments. Then, the comments identified by the model were re-identified by manual recognition. Finally, we used the official Weibo account “Psychological Map Psy” and sent a private message to the Weibo users marked as having a suicide risk. The contents of the private message mainly include inquiries about the user’s emotional state, expressions of concern, links to self-assessment psychological scales, and the hotline of the psychological crisis intervention center, etc. (Figure 3). In addition, users can choose whether to communicate with the voluntary counselor online according to their own wishes, so as to achieve agile psychological crisis intervention.

### 2.3. Data Collection and Processing

#### 2.3.1. Labeling the Interview Records Data

During the online psychological assistance project period (November 2016 to April 2019), the official Weibo account had communicated with 3627 users online, and 127,336 message records were accumulated. Adhering to the principle of privacy and confidentiality, the use of interview records data was strictly restricted. The experimenter must protect the privacy of the research subjects involved in the project, and should not use any confidential information related to the research subjects (including but not limited to accounts, nicknames, contact information, personal homepage, network IP, location, information records, etc.) outside the scope of the research, and would not violate the privacy of the subjects in any way.

The interview records data were encoded by qualitative analysis. The basic content of interview records data were the current status and problems of users. We extracted three indicators from the interview records data, and the specific descriptions of them are shown below:(1)Determining whether the user was in adolescent stages (junior high school, high school, college), and retaining the data labeled as the adolescents.(2)Determining whether the user was diagnosed with mental illness by a psychiatrist, had a history of mental illness, or was taking medication for mental illness, and labeling a user had a mental illness or not as a 0/1 dummy variable.(3)Coding the suicide stages (depressed mood, suicide ideation, suicide plan, suicide attempt) of adolescents, and marking the suicide stages as 1, 2, 3 and 4, an ordered categorical variable.

Suicide stages were coded on the basis of the following criteria: (a) Depressed mood: feeling negative and depressed, but not yet having suicidal ideation. It means participants did not express suicidal intention explicitly or implicitly(whether they actively expressed suicidal intentions, or exposed suicidal intentions under inquiries from the official Weibo account). (b) Suicide ideation: there is suicidal intent, but suicide plan has not formed. Due to being affected by an intense depressive mood, the idea of avoiding pain through suicide arises, but there is no specific time and way to implement it. (c) Suicide plan: on the basis of suicidal ideation, there is a rough or detailed suicide plan, such as when, where, and how to commit suicide. (d) Suicide attempt: at least one suicide attempt with the aim of ending one’s own life.

#### 2.3.2. Getting the Weibo Text Data

We scraped Weibo texts published by the same 3627 users through the Application Programming Interface (API) of Sina Weibo. In order to obtain sufficient Weibo texts for analysis, we took the first time that the official Weibo account sent a private message to the user as the beginning of suicide intervention, downloaded all Weibo texts in the 3 months before the intervention, and filtered the users whose Weibo texts are less than 300 words (Figure 4).

Finally, 105 high-suicide-risk adolescents (as the experimental group) were included in the analysis. We then removed stop words and performed Chinese word segmentation on the Weibo texts, and calculated word frequency. “The Chinese Suicide Dictionary” [27] was used to measure the word frequency ratio of their psychological pain and hopelessness. The dictionary outline is shown in Table 1. The psychological pain dimension represents the pain and torment of individuals at the psychological level, including 403 keywords; the hopelessness dimension represents the subjective feeling of losing hope, including 188 keywords.

### 2.4. Data Analysis

#### 2.4.1. Qualitative Analysis

Through the labeling results of interview records data, we investigated the situation of mental illness and the distribution of suicide stages for the experimental group.

#### 2.4.2. Analysis of Suicide Path

The mediation effect analysis was executed on the software Mplus by using the bootstrap method to test the suicide mechanism path model. The word frequency of psychological pain and hopelessness of the experimental group were treated as independent variable (X) and mediating variable (M), respectively. The suicide stages were treated as dependent variable (Y). The mental illness was included in the model as a control variable (Z).

## 3. Results

### 3.1. Group Characteristics Reflected by Qualitative Analysis

The mental illness diagnosis of the experimental group was marked; it was found that 38 were diagnosed and 67 were undiagnosed. The proportion of users diagnosed with mental illness was 36.2%. Among the 38 high-suicide-risk adolescents diagnosed with mental illness, 29 users had depression (or depression was included in multiple mental illnesses), accounting for 76.3%; the rest included bipolar disorder (13.16%), anxiety disorder (13.16%), etc.; 9 of them suffered from multiple mental illnesses (e.g., depression and anxiety disorder), accounting for 23.68%.

Among the experimental group, 57 were in the stage of depressed mood (54.3%), 21 were in the stage of suicide ideation (20.0%), 19 were in the stage of suicide plan (18.1%), and 8 people were in the stage of suicide attempt (7.6%) (Table 2).

### 3.2. Path Analysis of Psychological Pain, Hopelessness and Suicide Stages

A mediation effect test was performed on the path model (psychological pain → hopelessness → suicide stages). The performance of each fit index of the model was good (RMSEA < 0.08; CFI > 0.9; TLI > 0.9), indicating that the model fitting effect is excellent (Table 3).

After controlling for mental illness factors, the total effect c was significant (β = −0.154, *p* < 0.05), indicating that there was a mediation effect; the path coefficients a and b were significant (β = 0.571, *p* < 0.001; β = −0.271, *p* < 0.05), but the direct effect c’ was not significant (β = 0.052, *p* > 0.05), indicating that there was a complete mediating effect of hopelessness. That is, psychological pain mainly affects suicide risk through hopelessness (Table 4).

It was worth noting that the relationship between the expression of hopelessness and the stages of suicide showed a significant negative correlation (β = −0.271, *p* < 0.05), indicating that the late suicide stage such as suicide attempt, which may be closer to suicide death, included less expressions of hopelessness (Table 4).

Furthermore, there was a significant positive correlation between diagnosed mental illness and suicide stages (β = 0.398, *p* < 0.001), indicating the necessity of treating mental illness as a control variable in order to see the relationship between psychological pain, hopelessness and suicide stages more clearly. At the same time, it also showed that the later the suicide stage, the higher the diagnosis rate of mental illness (Table 4).

## 4. Discussion

### 4.1. The Qualitative Analysis Results

Among the experimental group, 36.2% of users suffered from mental illness. Depression accounted for 76.3% of all mental illness; it is the most common mental illness among the high-suicide-risk adolescents. The incidence of suicide among students whose relatives were affected by psychosis were higher than that of general residents. Suicide and mental illness were closely linked [28]. British doctor Sainsbury conducted a qualitative study on the data of 400 suicides examined by a coroner in London between 1936 and 1938, and found that 37% of the suicides reflected mental illness [29]. In a meta-analysis conducted in 2004, twenty-seven studies comprising 3275 suicides were included, of which 87.3% had been diagnosed with a mental disorder prior to their death [30]. In contrast, the research population in our study was only a high suicide risk group, so the detection rate of mental illness was relatively low. Although the research populations of the other two studies were both suicide decedents, the time interval between the two studies was large. It is very possible that there existed better methods to detect the mental illness of the suicide decedents in later studies, and it is understandable that the detection rate of mental illness has greatly increased. Furthermore, a more recent study has shown that the most common mental illness associated with suicide was depression [31]. The proportion of depression in all mental illnesses in our study also supported this conclusion.

Another result of the qualitative analysis showed that the later the suicide stage, the lower the proportion of adolescents. A study on residents’ suicidal ideation, suicide plans and suicide attempts in Dalian, China showed that the detection rate of suicidal ideation, suicide plan and suicide attempt decreased in that order in different groups [32]. This may because not all suicide ideations translate into suicidal actions, and most people with suicide ideation have not attempted suicide [33,34].

### 4.2. The Mediating Effect of Hopelessness

Our research found that the effect of psychological pain on suicide stages was totally mediated by hopelessness, confirming our suicide mechanism path hypothesis: psychological pain → hopelessness → suicide stages. Although the first step of suicide ideation starts from psychological pain, psychological pain itself is not enough to generate suicide ideation, and hopelessness is also needed to develop suicide ideation [19]. People make decisions and take actions based in part on their emotional predictions [35,36], and hopelessness makes people at suicide risk overestimate their future emotional pain [37]. If the individual experiences constant pain in life, this may reduce the individual’s desire to live, leading to suicidal thoughts. However, pain alone is not enough to completely lose the belief in life. Only when the individual feels that the terrible situation cannot be improved, and he or she will live in pain forever, suicide will be considered.

Thus, expressions of hopelessness may be a better predictor of suicide risk than psychological pain, and we found a traceable pathway for suicide intervention. Based on the Weibo posts, we can capture whether individual released suicide signals and it is possible to track the suicide stages in the future, making suicide intervention more accurate.

### 4.3. The Hopelessness Expressions and the Suicide Stages

There was a negative predictive relationship between hopelessness expressions and suicide stages, as adolescents at the later suicide stages had less expressions of hopelessness. One possible reason is that adolescents at higher risk of suicide may use less concrete, direct words of hopelessness to express such feelings. From a linguistic point of view, Gao Yihong and Meng Ling analyzed the Weibo texts of “Zoufan” three months before suicide. They found that “Zoufan” often used decontextualized and fragmented ways to express personal depressed mood, and expressed metaphors and intentions about death frequently [38]. This euphemistic and poetic way of venting may be the commonality of high suicide risk adolescents. Another possible reason is “the Presuicidal Syndrome”, which mentioned that there exists an “ominous quiet” in the short period before suicide. This is a sign of “affective restriction”, which represents extreme repression of intolerable feelings [39]. Such “affective restriction” may be why people in the later suicide stage are less likely to express hopelessness. Moreover, adolescents at the later suicide stage may be more accepting and tolerant of psychological pain and hopelessness, which makes them less willing to talk about individual psychological pain and hopelessness, and with lack of supportive survival signals through texts, lose the belief in getting help.

Many researchers have paid attention to the use of social media for suicide prevention and intervention, and have conducted a series of studies [40,41]. However, little attention has been paid to whether the suicidal signal was from an early suicide ideator or a later suicide attempter. Our study fills this gap, complementing and extending suicide-related research. Our results suggest that adolescents who are at a later suicide stage and have a higher suicide risk often do not express too much hopelessness directly. Therefore, we cannot judge closeness to suicide only by the level of traditional expressions of hopelessness.

## 5. Limitations and Future Studies

Although this study could divide the suicide stages of adolescents through labeling Weibo private message data, it cannot effectively determine the gender of the user. Previous studies had shown that suicidal behaviors were also affected by gender, and that the incidence of suicide attempts in adolescent females was higher than that of males [42]. Compared with males, females were more psychologically sensitive and vulnerable [43], and were more likely to express their pain through suicidal behaviors [44]. The expressions of hopelessness and psychological pain as predictors of suicide may differ by gender. There were only 105 adolescent users with high suicide risk in our study. Caution is thus necessary when generalizing the conclusions to other case.

Our findings provided a traceable pathway for suicide intervention, finding that the effect of psychological pain on suicide stages was totally mediated by hopelessness, and that adolescents in the later suicide stages had less expressions of hopelessness in the traditional sense. However, the specific language characteristics of each suicide stage are still unknown. If we want to accomplish the precise prediction of suicide stages through the expressions on Weibo posts and provide more accurate and effective information for suicide intervention, we need to further explore the specific language expression characteristics of high suicide risk adolescents in each suicide stage, especially how to identify abstract metaphors that may constitute a suicidal signal. It is thus possible that we can provide more accurate suicide intervention for adolescents at different suicide stages in the future.

We mainly focus on the expression on social media of two psychological indicators (psychological pain, hopelessness). However, there are many factors which affect suicide, and we can research suicide intervention from multiple perspectives, not only examining some negative indicators, but also exploring some positive indicators in future research. For example, previous studies had shown that social support [45], psychological resilience [46], and self-compassion [47] were protective factors for adolescent suicide.

## 6. Conclusions

The qualitative findings suggested that mental illness in high-suicide-risk adolescents was strongly associated with suicide risk, and that depression was the most common mental illness associated with suicide. Another finding was that the later the suicide stage, the smaller was the proportion of high-suicide-risk adolescents.

The results of the mediation effect analysis verified the suicide mechanism path: psychological pain → hopelessness → suicide stages, indicating that psychological pain mainly affected suicide stages through hopelessness, and hopelessness may be a better predictor of suicide risk for high-suicide-risk adolescents.

Notably, although adolescents in the later suicide stages were at greater suicide risk, they had less expression of hopelessness in the traditional sense.

## Figures and Tables

**Figure 1 ijerph-19-11495-f001:**
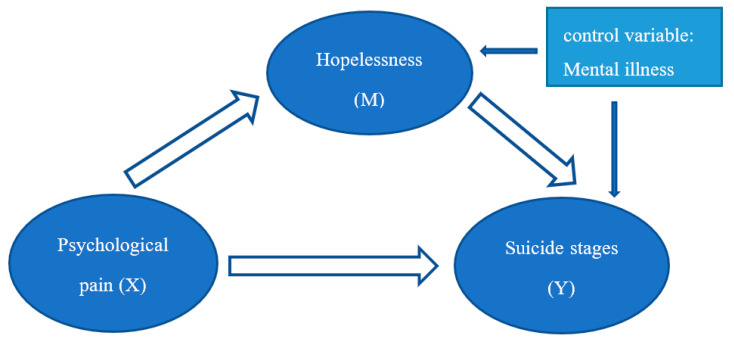
The mediation effect path model.

**Figure 2 ijerph-19-11495-f002:**
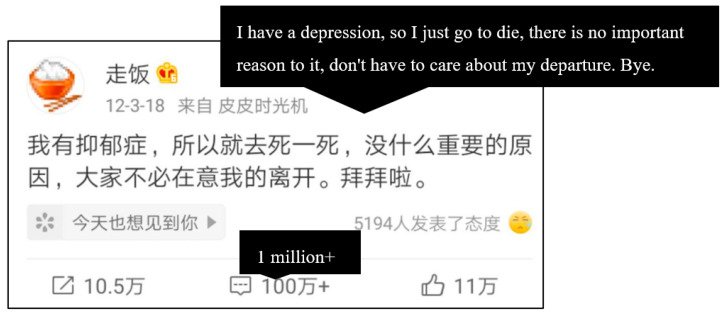
The last Weibo post of Weibo user “Zoufan”.

**Figure 3 ijerph-19-11495-f003:**
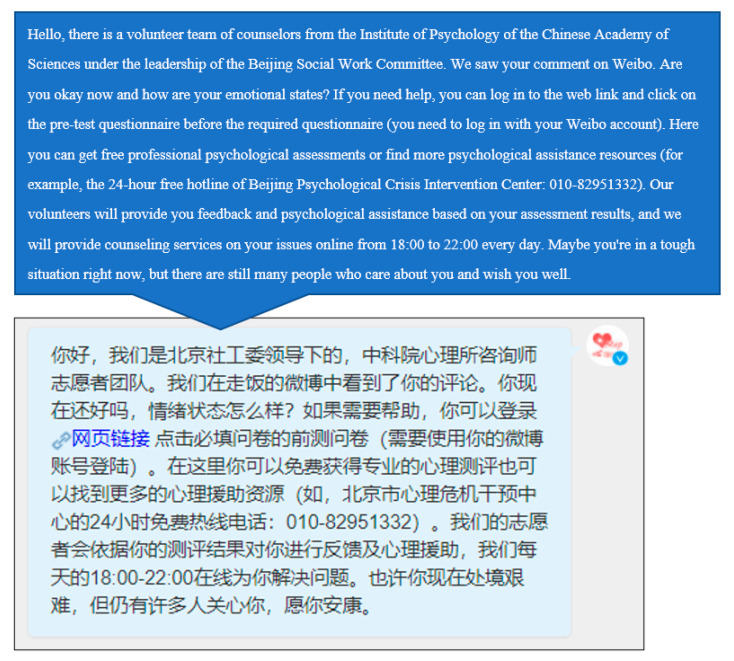
The first private message sent by the Weibo official account.

**Figure 4 ijerph-19-11495-f004:**
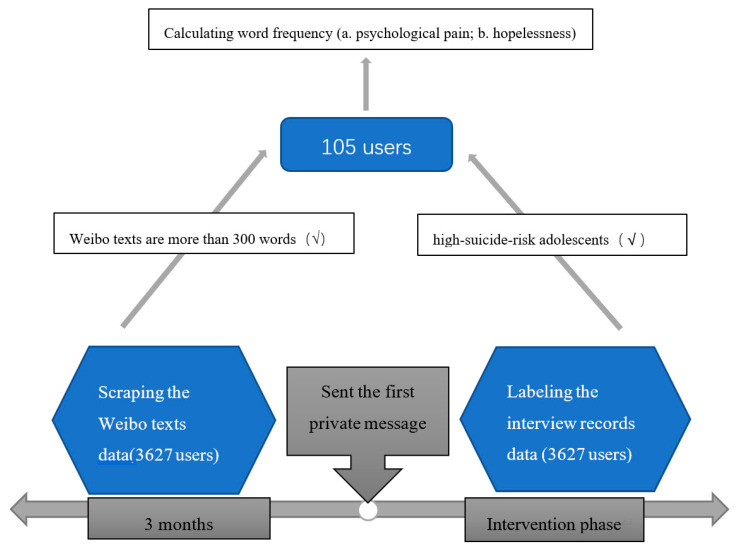
Data acquisition process.

**Table 1 ijerph-19-11495-t001:** Chinese suicide dictionary outline.

Category	Definition	Number of words	Representative words
Suicide ideation	Words reflecting suicidal thoughts	586	want to die (想死) escape (逃离)
Suicide behavior	Words reflecting self-harm behaviors	88	jump down (跳楼) hang(吊死)
Psychological pain	Words reflecting psychological distress	403	want to cry (想哭) loneliness (孤单)
Mental illness	Words reflecting poor mental health status	48	depression (抑郁) hallucination (幻觉)
Hopelessness	Words reflecting a feeling of despair	188	dead end (死胡同) despair (绝望)
Somatic complaints	Words reflecting somatic symptoms	183	headache (头疼) shortness of breath (透不过气)
Self-regulation	Words reflecting an attempt to push oneself hard	36	repression (压抑) force oneself to smile (强颜欢笑)
Personality	Words reflecting negative personality	72	inferiority complex (自卑) hate oneself (讨厌自己)
Stress	Words reflecting pressure in daily life	83	failure (输) pressure (压力)
Trauma/hurt	Words reflecting traumatic or unpleasant experiences	182	get dumped (失恋) infidelity (出轨)
Talk about others	Words reflecting one’s relatives and friends	47	partner (妻子) son (儿子)
Shame/guilt	Words reflecting a feeling of shame and guilt	72	lose status (丢脸) making an apology (赔罪)
Anger/hostility	Words reflecting a feeling of being angry and hostile against others	180	damn it (他妈的) curse (诅咒)

**Table 2 ijerph-19-11495-t002:** The distribution of suicide stages among the experimental group.

Depression Mood	Suicide Ideation	Suicide Plan	Suicide Attempt
Number (%)	Number (%)	Number (%)	Number (%)
57 (54.3%)	21 (20.0%)	19 (18.1%)	8 (7.6%)

**Table 3 ijerph-19-11495-t003:** The model fitting effect of path analysis.

RMSEA (<0.08)	CFI (>0.9)	TLI (>0.9)
0.000	1.000	1.000

**Table 4 ijerph-19-11495-t004:** Bootstrap mediation effect test results (β).

**Path**	**Total Effect** **(c)**	**X-M** **(a)**	**M-Y** **(b)**	**Direct Effects** **(c’)**
Psychological pain → hopelessness → suicide stages	−0.154 *	0.571 ***	−0.271 *	0.052
Mental illness → hopelessness	0.006			
Mental illness → suicide stages	0.398 ***			

Note: Values are Standardization Estimates, * *p* < 0.05, *** *p* < 0.001.

## Data Availability

To protect the participants’ privacy, the original posts used for the analysis are not publicly available but from the corresponding author at a reasonable request.

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
