# Peer review of "Exploring the Suicide Mechanism Path of High-Suicide-Risk Adolescents—Based on Weibo Text Analysis"

_ijerph, 2022, doi:10.3390/ijerph191811495_

Round 1

Reviewer 2 Report

Thanks for sending the paper. I am not sure what the authors attempted in this paper. What I understand is that they applied the suicide pathways based on Weibo text analysis. The paper revealed already stabilized facts from previous studies. I strongly oppose the process of assigning psychiatric diagnoses based on  Weibo text analysis.  Finally, it stands that the paper revealed already known factors with questionable methods. I may suggest that authors may consider contrasting the facts between Weibo data and clinical diagnoses. 

Round 2

Reviewer 1 Report

The authors have made a thorough review according to my suggestions. Two points were adequately addressed despite they were not exactly as I suggested, as good as possible.

I now recommend publication in its present form.

Sincerely,

Reviewer 2 Report

Thanks